# Evaluation of Autof MS2600 and MBT Smart MALDI-TOF MS Systems for Routine Identification of Clinical Bacteria and Yeasts

**DOI:** 10.3390/microorganisms12020382

**Published:** 2024-02-13

**Authors:** Elena De Carolis, Vittorio Ivagnes, Carlotta Magrì, Benedetta Falasca, Teresa Spanu, Maurizio Sanguinetti

**Affiliations:** Dipartimento di Scienze di Laboratorio e Infettivologiche, Fondazione Policlinico Universitario “A. Gemelli” IRCCS, 00168 Rome, Italy; vittorio.ivagnes@gmail.com (V.I.); carlottamagri97@gmail.com (C.M.); benedettafala97@outlook.it (B.F.); t.spanu@gmail.com (T.S.)

**Keywords:** clinical microbial identification, MALDI-TOF MS, evaluation Autof MS2600

## Abstract

The identification of microorganisms at the species level has always constituted a diagnostic challenge for clinical microbiology laboratories. The aim of the present study has been the evaluation in a real-time assay of the performance of Autobio in comparison with the Bruker mass spectrometry system for the identification of bacteria and yeasts. A total of 535 bacteria and yeast were tested in parallel with the two systems by direct smear or fast formic acid extraction for bacteria and yeasts, respectively. Discordant results were verified by 16S, ITS rRNA or specific gene sequencing. Beyond giving comparable results for bacteria with respect to the MBT smart system, Autof MS2600 mass spectrometer provided excellent accuracy for the identification of yeast species of clinical interest.

## 1. Introduction

Since 2008, the application of MALDI-TOF mass spectrometry technology in the microbiology laboratory has revolutionized the landscape of clinical microbial identification. Nowadays, this tool is widely used due to its accuracy, time to result, reduced reagent costs and thanks to its impact on the improvement of the diagnosis and treatment of infectious diseases. Different mass spectrometers are now available on the market, and among others, Autobio Diagnostics has recently introduced a new mass spectrometer: Autof MS2600. As widely reported by the scientific community, the database composition and its implementation through a constant update of mass profiles belonging to the bacterial and yeast species encountered in the clinical landscape is of detrimental importance [1]. This is particularly imperative when trying to identify rare species and genera poorly represented or missing in the mass spectrometer libraries. An additional variable for the optimization of the identification process starting from pure isolates cultured on solid medium relies on the preparation method used for the protein spectra generation prior to MALDI-TOF MS analysis. Specifically, on-plate direct transfer of the bacterial colonies on the target plate has been widely used for the identification of bacterial species of clinical interest with good results in terms of correct identification at the species level [2]. On the contrary, due to the intrinsic biological properties of the yeast’s cells relative to their hard cell wall, the identification of fungal pathogens has always been challenging [3]. To this extent, different extraction procedures have been applied to yeast colonies in order to obtain a high score of identification with variable results. As an alternative option for the accurate identification at the species level, the modification of the confidence score cut-off value has been applied, especially for what concerns on-plate direct transfer of yeast colonies [4]. The Bruker Biotyper algorithm uses a score ≥ 2 for secure identification at the species level, from 1.7 to 1.99 for genus only and below 1.7 for unreliable identification. Anyway, regarding the identification of clinical yeast isolates using the on-plate formic acid extraction method, by reducing the log score species threshold to 1.9 or 1.7, the success rate for yeast species identification increases [5]. Thanks to this lowered score value, a rise in correct identification percentage has been obtained further confirmed by sequencing analysis. Actually, this strategy, known as fast formic acid extraction, is commonly used in the microbiology clinical laboratories coupling higher speed and good identification results in comparison with the long extraction method [6].

Moreover, several studies tried to overcome the issue of confidence score cut-off value by the development of in-house-updated databases in order to improve the identification capability of MALDI-TOF MS for diagnosing clinical, emerging or rare isolates [7,8,9].

Concerning Autobio identification score intervals, matching results with score values above 9.0 are credible at the species level, from 6.0 to 8.99 at genus level and below 6.0 accounts for unreliable results.

Until now, only three papers have been published that referred to the Autobio MS1000 platform performance of identification [10,11,12,13]; nonetheless, to the best of our knowledge, there does not exist any research in the literature regarding the evaluation of the Autobio MS2600 mass spectrometer for the routine identification of clinical microorganisms.

Considering all the variables listed above, the aim of the present study has been the evaluation of the new Autof MS2600 (Autobio Diagnostics, Zhengzhou, China), in comparison with the MBT smart (Bruker Daltonics, Bremen, Germany) mass spectrometer by a real-time identification of bacteria and yeast retrieved from the clinical routine workflow or from an iced collection of samples covering the majority of microorganism species isolated in clinical microbiology laboratories.

## 2. Materials and Methods

### 2.1. Bacterial and Yeast Strains

The study relies on bacteria and yeast isolates collected at Fondazione Policlinico Gemelli IRCCS microbiology lab in Rome, Italy, during the routine workflow in 2023 or from our clinical isolates collection (22.5%) stored at −80 °C and sub cultured before testing. In particular, isolates belonging to those species that are challenging to identify due to close relatedness to other bacterial species or rarely isolated but clinically relevant (anaerobes and *Streptococci* or rare and emerging yeasts) were selected from our collection. Regarding colonies tested in real time, they were collected from blood (21%), pus and intra-abdominal (33.2%), respiratory (4.3%) and urine (19%) specimens. Concerning bacterial species, 38 genera and 80 species were tested, and concomitantly, regarding yeast isolates, 6 genera and 18 species were included in the analysis (Table 1).

A total of 535 isolates (434 bacteria and 101 yeasts) were analyzed in parallel using the Bruker Daltonics MBT smart and the Autobio Diagnostics Autof MS2600 mass spectrometers according to the Clinical and Laboratory Standards Institute (CLSI) M52 standard [14] by automatic matching of spectra profiles with the original database for Autobio identification system and original or original plus in-house-updated database [7] for what concerns the Bruker Biotyper identification system, for bacteria and yeasts, respectively. Isolates providing not reliable or discordant results were sequenced by 16S or specific genes in the case of bacteria and rRNA Internal Transcribed Spacer ITS 1–4 for yeasts [15]. In particular, GDH1/2, SO3 and SODA were used for *Streptococci* identification [16,17].

Bacterial DNA extraction was performed by High Pure PCR Template Preparation Kit following the manufacturer instructions (Roche Applied Science, Penzberg, Germany).

Concerning yeast DNA extraction, colonies from Sabouraud dextrose agar plate or Candida bromocresol green (BCG) (Vacutest Kima S.r.l., Arzergrande, Italy) were suspended in 500 μL of a glass beads (Sigma-Aldrich, St. Louis, MO, USA) sterile deionized water solution. After incubation at 95 °C for 30 min, the sample was frozen at −20 °C and fungal cells were disrupted by a Mini-beadbeater™ (BIOSPEC PRODUCTS, Bartlesville, OK, USA) in five steps of 10 s each. Subsequently to centrifugation at 15,000 rpm for five minutes, the supernatant was transferred in a new Eppendorf tube and DNA extraction was performed with DNeasy Plant Mini Kit (Qiagen, Hilden, Germany) following manufacturer instructions. PCR was performed with 100 ng of DNA using Hotstart Taq Master Mix Kit (Qiagen, Hilden, Germany).

Sequences were matched against the GenBank database and a BLAST software identification percentage above 98 was considered for gene sequence identification at the species level (http://www.ncbi.nlm.nih.gov/BLAST/ accessed on 18 December 2023) [18].

### 2.2. MALDI-TOF MS Identification

All identifications were performed starting from colonies grown at 37 °C on Blood Agar (TSA with 5% Sheep Blood), chocolate agar (PVX) or MacConkey Agar (bioMérieux, Grassina, Italy) for Gram-positive or Gram-negative bacteria, respectively, and Sabouraud Dextrose Agar (SDA) or Candida bromocresol green (BCG) (Vacutest Kima S.r.l.) for yeast isolates. The acquisition and matching of protein spectra, analyzed in duplicate in automatic mode from 240 shots in 40-shot steps by MALDI Biotyper (Bruker Daltonics) software package V4.1.14 was performed using the MBT Smart MALDI-TOF MS instrument in positive ion-mode with a laser frequency of 200 Hz. Instrument calibration was performed by Bruker Bacterial Test Standard (BTS). Spectra profiles were analyzed against the Bruker Biotyper V.11.0.0 library alone (covering 3893 species and 10.833 entries) for bacterial identification, whereas the updated yeast UCSC library [7] or the extended yeast (UCSC) and Bruker library combined were applied for yeast identification.

A log score ≥ 2.0 and ≥1.7 was recorded as correct identification to the species or to the genus level, respectively, and <1.7 for no reliable identification. An additional log score ≥ 1.9 was further evaluated. The best match reported for each duplicate spot analyzed was considered for the assay evaluation (Appendix A for bacteria and yeasts, respectively).

Following the Bruker identification, the same steel MSP 96 target plate was used to acquire in parallel the bacteria and yeast mass spectra profiles using the Autof MS2600 mass spectrometer by the Autof acquirer software package V2.0.196 and to match the profiles against the Autobio library V1120, covering 5189 species and 17,800 entries. For each spectrum, 240 shots in 40-shot steps from different positions of the target spot were acquired in automatic mode. The instrument is equipped with a 355 nm solid-state laser, and all the acquisitions were performed at 100 Hz in positive ion-mode.

The Autof MS calibrator based on nine typical peptide and proteins peaks relative to ribonuclease, myoglobin and proteins extracted from *E. coli* was used for the Autof MS2600 calibration as indicated by the manufacturer.

The manufacturer interpretation criteria were applied for samples profile matching; a log score value ≥ 9.0 for secure species identification, from 6.0 to 8.99 for identification at the genus level, and <7 for no reliable identification.

For all the measurements, the preparation method routinely performed during the workflow of the microbiology laboratory was followed according to the manufacturers’ instructions. In particular, isolates were prepared using direct smear on the target plate for bacterial colonies, as previously described [19]. For yeast isolates, cells from a single colony were subjected to a fast extraction procedure by on-plate formic acid treatment by adding the formic acid solution prior to colony transfer [3]. One microliter of α-cyano-4-hydroxycinnamic acid matrix (CHCA) was added onto the samples and let air-dry before the introduction in the mass spectrometer.

## 3. Results 

Regarding the 434 bacterial isolates tested, the percentage of correct identification using MBT smart system was 98.39 and 98.85 at a log score ≥ 2 or ≥1.9, respectively, and reached the totality of the samples examined at the log score of 1.7. On the other hand, Autof MS2600 correctly identified 99.31% of the bacterial isolates at a log score ≥ 9.0 and 100% of the bacterial isolates at a log score of ≥7.0 (Appendix A). In particular, the misidentifications were related to the following species for Bruker system: 1 *Acinetobacter baumannii* isolate out of 11 was misidentified as *A. nosocomialis,* 1 *Citrobacter freundiis* was incorrectly identified as *C. brakii* and 1 *Klebsiella pneumoniae* out of 35 was reported as *K. variicola*. With regard to the Autobio system, one *Enterobacter hormaechei* isolate out of four was incorrectly identified as *E. cloacae*. Considering bacterial species identified at the genus level only, one *E. coli* isolate was identified in the Bruker system with a log score of 1.83, and one *Streptococcus infantis* at 1.76. Moreover, no *S. infantis* isolate was identified by the Bruker system at the species level above 2.0, whilst one out of three *S. infantis* was correctly identified by the Autobio system. For the two remaining isolates, one was identified by the Bruker system at a log score of 1.76, whilst the Autobio report resulted in a *S. oralis* at a log score of 7.23. The last *S. infantis* isolate was misidentified by both systems (Appendix A).

Variable results were obtained with the two systems for the 101 identifications performed on yeast isolates (Appendix A). 

In particular, Autof MS2600 correctly classified 98.02% of all the yeast isolates with a reliable identification score ≥ 9.0, and the percentage reached 99.01 once considering a score ≥ 7.0. 

For the MBT smart system, conversely, we needed to apply different score values and combine the in-house-updated UCSC and Bruker library to reach a good percentage of correct identifications (96.04%). Specifically, identification at the species level raised from 38.61% to 85.15% once considering instrument reference library alone or in-house-updated yeast database at score value equal or above 2.0, from 45.55% to 93.07% using a score value ≥ 1.9 and from 61.39% to 96.04% considering a score ≥ 1.7 (Table 2).

The species misidentified by Bruker system applying the extended database consisted in: two *C. haemulonii* correctly classified by Autobio system, one *C. neoformans* out of five identified at the genus level (score 1.8) by Autobio, and the *S. clavata* isolate misidentified by both instruments.

In summary, MBT smart allowed the correct identification of 97.94%, 98.32% and 99.25% of the clinical microbial specimens once considering a score value ≥ 2, ≥1.9 or ≥1.7, respectively, whilst Autof MS2600 correctly identified 99.07% and 99.81% of the isolates at the species and at the genus level (identification score ≥ 9.0 or ≥7.0, respectively).

## 4. Discussion

Nowadays, thanks to its superior ability in terms of time to result and accuracy, MALDI-TOF mass spectrometry constitutes a fundamental method of identification at the species level for pathogenic microorganisms, thus replacing the biochemical and phenotypic analytical methods previously adopted in the routine workflow of clinical laboratories. 

Until now, the two major MALDI-TOF MS-based microbial identification systems used in the clinical microbiology laboratories have been Biotyper (Bruker Daltonics) and VITEK MS (bioMérieux) along with Shimadzu, but the field has grown in Asian countries in recent years, and new mass spectrometry instruments have been introduced in the market.

To this regard, the Autobio MS1000 mass spectrometer from Autobio Diagnostics has been recently presented as the first system in the landscape of the identification of clinical pathogenic microorganisms. The instrument is equipped with an ion source vacuum (up to 10^−7^ mPa) and possesses higher speed over the existing systems being able to deliver a full target plate (96 tests) in about 20 min; moreover, its performance of identification has been proved comparable to the commonly used MALDI-TOF systems [10].

On the wave of this success, the Autof MS2000 and Autof MS2600 MALDI-TOF Microbial ID Systems are now available on the market. 

To our knowledge, this paper constitutes the first evaluation dealing with the performance of identification on clinical microbial species using the new Autof MS2600 mass spectrometer. The findings of this study, in particular the superior ability in the identification of yeast isolates, can be of interest for the scientific community and could constitute an additional value, especially in specific clinical settings (e.g., hematology, intensive care) in which timely diagnosis can make the difference for the patient outcome. 

As reported in several papers, the implementation of the software database with additional spectra acquired from rare or difficult-to-identify species can be of detrimental importance in order to achieve high scores value and secure identification at the species level. 

Moreover, beyond commercial or in-house-developed databases, one of the last resources for MALDI-TOF MS identifications consists of free available on-line databases as “Microbenet” or “MSI” (https://microbenet.cdc.gov; https://msi.happy-dev.fr/ accessed on 18 December 2023) able to improve the identification of challenging, rare or emerging strains at the species level.

Regarding bacterial species such as *Escherichia coli*, *Propionibacterium acnes*, *Shigella* spp., some strains of *Stenotrophomonas maltophilia*, or *Streptococcus pneumoniae*, and members of the *S. oralis*/*mitis* group, it is historically known that they can be misidentified by MALDI-TOF MS due to the low rate of differences in their ribosomal protein sequences. Anyway, the latest version of the Bruker Biotyper database 4.1 has shown better performance in identifying nonpneumococcal viridans group *streptococci* (VGS) as compared to the previous one (v. 3.1), although the *S. mitis* and *S. bovis* group would need further specific gene sequencing for correct identification [20]. 

In this study, forty-nine isolates belonging to *Streptococcus* genera were tested in parallel with Bruker and Autobio systems. According to the previous literature, misidentifications were reported for *S. mitis* group; in fact, we observed for *S. infantis* no correct identification at the species level for Bruker and one correct identification at the species level for Autobio system (Appendix A). Anyway, misidentification as pneumococci was never observed in the *S. mitis* group of isolates tested in parallel with the two systems.

One additional observation is that Biotyper system still retains the *S. mitis_oralis* taxonomy in the identification report, whilst the Autobio system matches the two species as distinct ones; anyway, the six *S. mitis_oralis* isolates were correctly identified by Autobio system as *S. mitis* or *S. oralis*.

The possibility of fast and accurate identification between *S. pneumonia* from non-pneumococcal streptococci by MALDI-TOF MS is of detrimental importance when also considering that discrimination among the *S. mitis* group is challenging even using molecular methods as notoriously known and as experimented in this study, requiring different DNA probes and sequencing. Given the importance from a clinical point of view of a rapid detection of bacteria being normal commensal of oral cavity or causing severe infections as endocarditis, the reduction in the time to diagnosis is a key step to start an effective therapy and improve clinical outcomes.

For what concerns yeast species of medical importance, the strategy for an accurate identification at the species level has traditionally been a challenge which relies on different identification score algorithms, sample pretreatment procedures or the validation of extended database including rare or difficult-to-identify species [3]. In particular, if one considers the menace of emerging clinically relevant yeast species (*C. auris*), and *Candida* cryptic species (*C. parapsilosis complex*, *C. glabrata complex*), considering that the epidemiologic scenario is moving towards an increased prevalence of non-*C. albicans* species, it is evident that the accurate identification of fungi at the species level is of paramount importance to start a surveillance strategy and to prevent outbreaks related to multi-drug-resistant species.

Notwithstanding these circumstances, not only accurate but early identification of species in clinical setting is detrimental to prevent outbreaks related to multi-drug-resistant species and primarily for optimal patient clinical management. Moreover, the importance of timely identification at the species level is evident especially once considering that the antifungal susceptibility of *Candida* species is species-specific; therefore, in this context, MALDI-TOF MS offers good performance and can reduce the time to appropriate and effective therapy. 

However, also in the case of the Autof system, users have the option to develop their own database, and the implementation of the number of existing profiles has not been necessary due to the high performance and accuracy of results reached for yeast identification using the manufacturer-construct library only.

Considering the experience obtained from our previous studies [7], having implemented and constantly updated the existing Biotyper database with additional profiles of 48 genera and 11 species of yeasts, we can state that the performance obtained by the Autof MS2600 system regarding yeast samples is astounding. Moreover, it was not necessary to lower the identification threshold as a last resort as the actual score of identification above 9.0, indicated by the manufacturer, has been evaluated as sufficient to obtain more than 98% of correct yeast identification at the species level.

One of the possible explanations for this success in the identification at the species level, especially for yeasts, might be found in the amount of entries included in the Autof database; 17,800 strains, 5189 species, 1064 genera (Table 3). In fact, it is notoriously known in the mass spectrometry landscape that the library dimension is one of the key points for a successful identification rate of microorganisms, especially in the case of rare or emerging pathogens [21,22].

Moreover, considering the linear flight tube drift length being 10 cm longer for Autof MS2600 with respect to the MBTsmart mass spectrometer and mass accuracy being higher (Table 3), we hypothesize that some technical aspects along with the amount of entries could have an influence on the performance of identification at the species level, especially for what concerns yeast isolates. In fact, an increase in the number of ribosomal marker peaks detected and a decrease in the mass measurement error has been argued to have an influence in spectra quality which correlates with correct identification at the species level [2]. Anyway, further studies would be required to assess this question.

In conclusion, basing on the results obtained by the comparison with the MBT smart instrument and sequencing analysis of the discrepant or unidentified species, Autof MS2600 is equivalent to the Bruker system with regard to bacteria of clinical interest, reaching 99.31% vs. 98.39% of correct identification at the species level. Moreover, its performance is outstanding if one considers the percentage of secure identification at the species level of *Candida* and rare or emerging yeasts (98.02%) achieved by spectra profiles matching against the Autobio system library only.

Beyond the evident clinical impact, the optimal accuracy obtained in identifying clinical bacteria and yeasts led to a positive impact on the routine laboratory workflow allowing us to save time and reduce costs for repetitions of samples measurements, multi-step extractions protocols, DNA sequencing or acquisition of multiple spectra profiles needed for the database implementation. Anyway, for some microorganisms (e.g., *Saprochaete clavata*, *S. mitis* group), identification at the species level actually constitutes a limitation for MALDI-TOF mass spectrometry, particularly for those species subjected to taxonomic changes; in this case, supplementary tests are still required to further confirm or determine the species’ exact identification.

We are conscious that rare bacteria or filamentous fungi were not included in this analysis dealing mainly with the species commonly isolated in the clinical microbiology laboratory; a subsequent evaluation will cover these pathogens in order to further analyze the power of identification of the new Autof MS2600 mass spectrometer.

Ethics Committee or Institutional Review Board approval was not required for this manuscript as the study involves anonymous blinded microorganism strains only, and no patient information is retrievable, nor it is included in the manuscript.

## Figures and Tables

**Table 1 microorganisms-12-00382-t001:** Summary of identification results for bacterial and yeast species tested by Autobio MS2600 and MBT smart systems.

Bacterial Species		Autobio MS2600			MBT Smart ^a^			
	n. Tested	≥9.0	≥7.0	Incorrect	≥2.0	≥1.9	≥1.7	Incorrect
*Achromobacter xylosoxidans*	2	2/2	-	-	2/2	-	-	-
*Acinetobacter baumannii*	11	11/11	-	-	10/1	-	-	1/10
*Actinomyces europaeus*	1	1/1	-	-	1/1	-	-	-
*Actinomyces neuii*	1	1/1	-	-	1/1	-	-	-
*Actinomyces turicensis*	2	2/2	-	-	2/2	-	-	-
*Aeromonas hydrophila*	1	1/1	-	-	1/1	-	-	-
*Bacteroides faecis*	2	2/2	-	-	2/2	-	-	-
*Bacteroides fluxus*	1	1/1	-	-	1/1	-	-	-
*Bacteroides fragilis*	8	8/8	-	-	8/8	-	-	-
*Bacteroides ovatus*	2	2/2	-	-	2/2	-	-	-
*Bacteroides thetaiotaomicron*	2	2/2	-	-	2/2	-	-	-
*Bifidobacterium bifidum*	1	1/1	-	-	1/1	-	-	-
*Burkholderia gladioli*	1	1/1	-	-	1/1	-	-	-
*Citrobacter freundii*	2	2/2	-	-	1/2	-	-	1/2
*Citrobacter koseri*	1	1/1	-	-	1/1	-	-	-
*Clostridium hathewayi*	1	1/1	-	-	1/1	-	-	-
*Clostridium perfringens*	1	1/1	-	-	1/1	-	-	-
*Clostridium ramosum*	1	1/1	-	-	1/1	-	-	-
*Clostridium septicum*	1	1/1	-	-	1/1	-	-	-
*Corynebacterium striatum*	5	5/5	-	-	5/5	-	-	-
*Cutibacterium acnes*	2	2/2	-	-	2/2	-	-	-
*Dermabacter hominis*	3	3/3	-	-	3/3	-	-	-
*Eggerthella lenta*	1	1/1	-	-	1/1	-	-	-
*Enterobacter cloacae*	4	4/4	-	-	4/4	-	-	-
*Enterobacter hormaechei*	4	3/4	-	1/4	4/4	-	-	-
*Enterobacter kobei*	4	4/4	-	-	4/4	-	-	-
*Enterococcus avium*	2	2/2	-	-	2/2	-	-	-
*Enterococcus faecalis*	31	31/31	-	-	31/31	-	-	-
*Enterococcus faecium*	25	25/25	-	-	25/25	-	-	-
*Enterococcus gallinarum*	3	3/3	-	-	3/3	-	-	-
*Enterococcus malodoratus*	1	1/1	-	-	1/1	-	-	-
*Escherichia coli*	41	41/41	-	-	40/41	-	1/41	-
*Haemophilus haemolyticus*	2	2/2	-	-	2/2	-	-	-
*Haemophilus influenzae*	4	4/4	-	-	4/4	-	-	-
*Hafnia alvei*	2	2/2	-	-	2/2	-	-	-
*Klebsiella aerogenes*	2	2/2	-	-	2/2	-	-	-
*Klebsiella oxytoca*	6	6/6	-	-	6/6	-	-	-
*Klebsiella pneumoniae*	35	35/35	-	-	34/35	-	-	1/35
*Klebsiella variicola*	1	1/1	-	-	1/1	-	-	-
*Lacticaseibacillus rhamnosus*	1	1/1	-	-	1/1	-	-	-
*Lactobacillus delbrueckii*	1	1/1	-	-	1/1	-	-	-
*Lactobacillus gasseri*	2	2/2	-	-	2/2	-	-	-
*Lactobacillus rhamnosus*	4	4/4	-	-	4/4	-	-	-
*Morganella morganii*	9	9/9	-	-	9/9	-	-	-
*Pantoea ananatis*	1	1/1	-	-	1/1	-	-	-
*Parvimonas micra*	1	1/1	-	-	1/1	-	-	-
*Peptostreptococcus anaerobius*	1	1/1	-	-	1/1	-	-	-
*Prevotella bivia*	2	2/2	-	-	2/2	-	-	-
*Proteus mirabilis*	25	25/25	-	-	25/25	-	-	-
*Providencia rettgeri*	1	1/1	-	-	1/1	-	-	-
*Providencia stuartii*	3	3/3	-	-	3/3	-	-	-
*Pseudomonas aeruginosa*	30	30/30	-	-	30/30	-	-	-
*Serratia marcescens*	2	2/2	-	-	2/2	-	-	-
*Staphylococcus aureus*	34	34/34	-	-	34/34	-	-	-
*Staphylococcus capitis*	6	6/6	-	-	6/6	-	-	-
*Staphylococcus caprae*	1	1/1	-	-	1/1	-	-	-
*Staphylococcus epidermidis*	24	24/24	-	-	24/24	-	-	-
*Staphylococcus haemolyticus*	7	7/7	-	-	7/7	-	-	-
*Staphylococcus hominis*	2	2/2	-	-	2/2	-	-	-
*Staphylococcus lugdunensis*	2	2/2	-	-	2/2	-	-	-
*Staphylococcus pasteuri*	1	1/1	-	-	1/1	-	-	-
*Staphylococcus pseudintermedius*	1	1/1	-	-	1/1	-	-	-
*Stenotrophomonas maltophilia*	2	2/2	-	-	2/2	-	-	-
*Streptococcus agalactiae*	9	9/9	-	-	9/9	-	-	-
*Streptococcus anginosus*	7	7/7	-	-	7/7	-	-	-
*Streptococcus constellatus*	1	1/1	-	-	1/1	-	-	-
*Streptococcus dysgalactiae*	2	2/2	-	-	2/2	-	-	-
*Streptococcus gallolyticus*	7	7/7	-	-	7/7	-	-	-
*Streptococcus gordonii*	1	1/1	-	-	1/1	-	-	-
*Streptococcus infantis*	3	1/3	1/3	1/3	0/3	-	2/3	1/3
*Streptococcus intermedius*	1	1/1	-	-	1/1	-	-	-
*Streptococcus oralis*	3	3/3	-	-	0/3	-	-	-
*Streptococcus mitis*	3	3/3	-	-	0/3	-	-	-
*Streptococcus parasanguinis*	1	1/1	-	-	1/1	-	-	-
*Streptococcus pneumoniae*	3	3/3	-	-	3/3	-	-	-
*Streptococcus pseudopneumoniae*	1	1/1	-	-	1/1	-	-	-
*Streptococcus pyogenes*	6	6/6	-	-	6/6	-	-	-
*Streptococcus salivarius*	1	1/1	-	-	1/1	-	-	-
*Veillonella atypica*	1	1/1	-	-	1/1	-	-	-
*Yersinia enterocolitica*	1	1/1	-	-	1/1	-	-	-
Total	434							
**Yeast Species**	**n. tested**	**Autobio MS2600**			**MBT smart ^b^**			
		**≥9.0**	**≥7.0**	**Incorrect**	**≥2.0**	**≥1.9**	**≥1.7**	**Incorrect**
*Candida albicans*	35	35/35	-	-	35/35	-	-	-
*Candida auris*	4	4/4	-	-	4/4	-	-	-
*Candida dubliniensis*	2	2/2	-	-	2/2	-	-	-
*Candida glabrata*	10	10/10	-	-	10/10	-	-	-
*Candida haemulonii*	2	2/2	-	-	0/2	-	-	2/2
*Candida kefyr*	1	1/1	-	-	1/1	-	-	-
*Candida krusei*	5	5/5	-	-	5/5	-	-	-
*Candida metapsilosis*	6	6/6	-	-	6/6	-	-	-
*Candida nivariensis*	2	2/2	-	-	2/2	-	-	-
*Candida norvegensis*	2	2/2	-	-	2/2	-	-	-
*Candida orthopsilosis*	5	5/5	-	-	5/5	-	-	-
*Candida parapsilosis*	4	4/4	-	-	4/4	-	-	-
*Candida tropicalis*	7	7/7	-	-	7/7	-	-	-
*Cryptococcus neoformans*	5	4/5	1/5	-	4/5	-	-	1/5
*Rhodotorula mucillaginosa*	5	5/5	-	-	5/5	-	-	-
*Saccharomyces cerevisiae*	3	3/3	-	-	3/3	-	-	-
*Saprochaete clavata*	1	0/1	-	1/1	0/1	-	-	1/1
*Trichosporon asahii*	2	2/2	-	-	2/2	-	-	-
Total	101							

^a^ Spectra matched against commercial Bruker and ^b^ commercial plus in-house-updated UCSC database, respectively.

**Table 2 microorganisms-12-00382-t002:** Percentage of correct identification for bacteria and yeasts.

	n. Correct Identification (%)			n. Undetermined (%)
	Species		Genus	
MBTsmart	≥2	≥1.9	≥1.7	
Bacteria	98.39	98.85	100	-
Yeasts	38.61	45.55	61.39	38.61
Yeasts updated	85.15	93.07	96.04	3.96
Yeasts combined	96.04	96.04	96.04	3.96
Total	97.94	98.32	99.25	0.75
Autof MS2600	≥9.0		≥7.0	
Bacteria	99.31		100	-
Yeasts	98.02		99.01	0.99
Total	99.07		99.81	0.19

**Table 3 microorganisms-12-00382-t003:** Comparison of technical aspects of Biotyper MBT smart and Autof MS2600.

Technical Specifications	Biotyper MBT Smart	Autof MS2600
Laser	337 nm solid-state	355 nm solid-state
Maximum pulse rate	200 Hz	300 Hz
Linear flight tube drift length	0.95 m	1.05 m
Mass analyzer	Linear TOF	Linear TOF
Mass accuracy	150 ppm	<100 ppm
Database	3893 species, 10.833 entries	5189 species, 17800 entries

## Data Availability

The data that supports the findings of this study are available in the Appendix A of this article.

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
