# Peer review of "Evaluation of Autof MS2600 and MBT Smart MALDI-TOF MS Systems for Routine Identification of Clinical Bacteria and Yeasts"

_microorganisms, 2024, doi:10.3390/microorganisms12020382_

Round 1
Reviewer 1 Report
Comments and Suggestions for Authors
The paper aims to evaluate the performance of the Autof MS2600 mass spectrometer in identifying bacterial and yeast isolates in clinical settings. The study contributes valuable insights into the comparison between Autof MS2600 and Bruker Daltonics MBT smart systems, emphasizing their accuracy in identifying clinically relevant microorganisms. The strength of the paper lies in its comprehensive analysis of a diverse set of isolates, including bacteria and yeasts, and its emphasis on the system's performance in identifying emerging species. The paper concludes that the Autof MS2600 is comparable to the Bruker system for bacterial identification and excels in yeast identification, presenting a potential advancement in clinical microbiology.
The strengths include the inclusion of a diverse set of isolates and the comparative analysis with an established system. However, the paper could benefit from a more detailed discussion on potential limitations and challenges faced during the evaluation. Additionally, clarity is needed regarding the selection criteria for bacterial and yeast isolates, and the paper should discuss any biases introduced by this selection. The relevance of the study is clear, especially considering the importance of accurate and timely microbial identification in clinical practice. However, the authors should highlight the potential impact of their findings on routine laboratory workflows.
Specific Comments:
- Lines 200 to 202 mentions the first evaluation of Autof MS2600, but it would be beneficial to include a brief discussion on how this study contributes to the existing literature on MALDI-TOF MS systems.
- The discussion on yeasts (lines 231-242) is insightful; however, more context on the clinical implications of accurate yeast identification would enhance the practical relevance of the findings.
- While the paper discusses the systems' performance with Streptococcus species, a clearer delineation of how this aligns with clinical implications and potential patient outcomes would strengthen the discussion.
Author Response
We are grateful to the reviewer for his appreciated comments that in our opinion have been fundamental for implementing the paper value and add interesting insights from a clinical point of view, some lines of the manuscript have been modified accordingly to the comments and highlighted in yellow.
In particular:
The potential limitations and challenges faced during the evaluation have been discussed further in the lines 295-303 as requested.
The selection criteria for bacteria and yeasts isolates have been explained more clearly in the lines 75-79.
The biases introduced by the selection can be found in lines 304-307.
The potential impact of the findings on routine laboratory workflow has been included in the lines 295-303.
Regarding the specific comments on how this study contributes to the existing literature, this topic has been further discussed in the lines 204-207.
The clinical implications of accurate yeasts identification have been added and highlighted in the lines 255-258 as indicated.
How the performance with Streptococcus species aligns with clinical implications has been discussed further in the lines 236-243 as suggested.
Reviewer 2 Report
Comments and Suggestions for Authors
The manuscript titled "Evaluation of Autof MS2600 and MBT smart MALDI-TOF MS systems for routine identification of clinical bacteria and yeasts" submitted for review in the journal Microorganisms describes a comparison of two MALDI MS systems used in microbial identification. The paper is of interest because of the great interest in MALDI systems for diagnostic laboratories and the ever emerging newly developed instruments and databases by various companies. An example of this is this work where a brand new instrument is compared with an instrument that is already well-established on the market. In my humble opinion, the work is well prepared and it is possible to accept it for publication in this journal with minor corrections.
1. I would ask you to add to the manuscript a table comparing the technical parameters of the two instruments.
2. I think it would be a good idea to present the results graphically, such as pie charts showing the percentage of strains correctly identified, and strains not identified on both systems. Also, Venn diagrams describing how many samples were classified identically or differently by both systems would be an interesting way to present the results.
3. Please try to explain in the manuscript where the differences in microbial identification between the two systems might come from.
Author Response
We thank the reviewer for the positive evaluation of this manuscript and for his comments that allowed us to better underline our findings. We added the minor corrections requested accordingly and we highlighted in yellow the modified parts of the manuscript.
In particular:
- Table 3 comparing the technical parameters of the two instruments has been added to the manuscript at line 277
- A graphical abstract has been added to the manuscript useful to present the results both as pie chart and venn diagram.
- The differences in microbial identifications between the two systems have been hypothesized and commented further in lines 280-287 as suggested.